# DNA barcoding echinoderms from the East Coast of South Africa. The challenge to maintain DNA data connected with taxonomy

**Gontran Sonet[1]\*, Nathalie Smitz[2], Carl Vangestel[1], Yves Samyn[3]**

**1** Joint Experimental Molecular Unit—JEMU, Operational Directorate Taxonomy and Phylogeny, Royal Belgian Institute of Natural Sciences, Brussels, Belgium, **2** Joint Experimental Molecular Unit—JEMU, Department of Biology, Royal Museum for Central Africa, Tervuren, Belgium, **3** Recent Invertebrates Collections, Scientific Heritage Service, Royal Belgian Institute of Natural Sciences, Brussels, Belgium

\* gsonet@naturalsciences.be

**Data Availability Statement:** All DNA sequences are available in the Barcode of Life Data Systems (BOLD) with Process IDs from CHARE001-20 to CHARE400-20 (project CHARE).

## Abstract

Echinoderms are marine water invertebrates that are represented by more than 7000 extant species, grouped in five classes and showing diverse morphologies (starfish, sea lilies, feather stars, sea urchins, sea cucumbers, brittle and basket stars). In an effort to further study their diversity, DNA barcodes (DNA fragments of the 5' end of the cytochrome c oxidase subunit I gene, COI) have been used to complement morphological examination in identifying evolutionary lineages. Although divergent clusters of COI sequences were reported to generally match morphological species delineations, they also revealed some discrepancies, suggesting overlooked species, ecophenotypic variation or multiple COI lineages within one species. Here, we sequenced COI fragments of 312 shallow-water echinoderms of the East Coast of South Africa (KwaZulu-Natal Province) and compared morphological identifications with species delimitations obtained with four methods that are exclusively based on COI sequences. We identified a total of 103 morphospecies including 18 that did not exactly match described species. We also report 46 COI sequences that showed large divergences (>5% p-distances) with those available to date and publish the first COI sequences for 30 species. Our analyses also identified discordances between morphological identifications and COI-based species delimitations for a considerable proportion of the morphospecies studied here (49/103). For most of them, further investigation is necessary to keep a sound connection between taxonomy and the growing importance of DNA-based research.

## Introduction

Echinoderms are exclusively marine water invertebrates whose larvae are bilaterally symmetric while the adults show the typical pentamerous radial symmetry, which can be doubled by a secondary bilateral symmetry (cf. holothuroids or irregular echinoids). Their phylum (Echinodermata) includes five extant classes: Asteroidea (starfish or sea stars), Crinoidea (sea lilies and

**Funding:** The DNA analysis performed in this study was financed by the Belgian Science Policy. Funding for this work also came from the Flemish Community Bilateral (International) Scientific and Technological Cooperation, project numbers BIL98/84 (YS) and BIL01/46 (YS), the Fund for Scientific Research Flanders (YS), the Research Council of the Free University Brussels (YS), the Belgian Global Taxonomic Initiative (YS) and the King Leopold III Fund for Nature Exploration and Conservation (YS). These funders had no role in study design, data collection and analysis, decision to publish, or manuscript redaction.

**Competing interests:** The authors have declared that no competing interests exist.

feather stars), Echinoidea (sea urchins), Holothuroidea (sea cucumbers) and Ophiuroidea (brittle and basket stars). They are represented by more than 7000 extant and 13000 extinct species [1] but this number likely is a gross underestimation given that taxa continue to be described annually [2–7]. In addition to morphology, a mitochondrial DNA region of the 5' end of the cytochrome *c* oxidase subunit I gene (COI) is widely used as a DNA barcode for the identification of animal species [8]. It has been used as a useful proxy to study the diversity of extant echinoderm species because clusters of similar COI sequences largely corroborated morphological identifications in several large-scale studies including all five classes of echinoderms from Canadian waters [9], the coasts of Australia and New Zealand [10] and the North Sea [11]. Yet, divergent clusters of COI sequences were observed within some species [9–12]. Some of these divergent clusters were assigned to undescribed species on the basis of detailed taxonomic analyses combining morphological characteristics, DNA data and sometimes ecological observation [13–17]. Divergent clusters of COI sequences were also observed within *Linckia multifora* (distances up to 0.023) and *L. laevigata* (divergent mitogenomes) [18, 19], but could not be associated with morphological differentiation, suggesting either cryptic speciation or hybridization. Similarly, an integrative taxonomical study on the ophiuroid *Astrotoma agassizii* suspected that seven well-supported and deeply divergent COI clades found in this group corresponded to cryptic species despite evidence of admixture [20, 21]. In contrast, divergent COI lineages (0.011–0.044 distances) were shown to coexist in two species of the *Ophioderma longicauda* species complex recently delimited based on morphology, developmental features, and mitochondrial and nuclear (coding and non-coding) DNA [22, 23]. These studies show that clusters of COI sequences can reveal lacunes of knowledge in echinoderm species diversity and that integrative taxonomy brings added value to address some species complexes.

In the Republic of South Africa, despite a relatively strong tradition of marine taxonomic research [24] and the availability of voucher specimens from historical and recent biodiversity inventories, the shallow-water echinoderms of the North-East coast of the country have until recently remained poorly documented. In addition, there is a poor representation of DNA barcodes from South African echinoderms in the online DNA repositories compared to other regions of the world. In fact, less than 500 public COI sequences are available for South African echinoderms out of more than 34000 public sequences in the Public Data Portal of BOLD [25]. In this context, we undertook the DNA barcoding of recently collected shallow-water echinoderms of the North-East coast of South Africa (KwaZulu-Natal Province). Our goals are to (i) provide DNA barcodes for recently (1999–2016) sampled shallow-water echinoderms from the North and South of the KwaZulu-Natal coast, (ii) compare morphological identifications with species delimitations exclusively based on COI sequences and (iii) flag specimens for which additional taxonomic investigations are needed. This exercise is useful to describe the species diversity of echinoderms in South Africa but also on earth, as South Africa is situated in the West of the Indian Ocean, which is known for its exceptional marine diversity [26].

## Materials and methods

### Sampling and morphospecies identification

A total of 312 specimens were collected during five different campaigns (from 1999 to 2016) in the shallow-waters North and South of the KwaZulu-Natal Province, on the East Coast of the Republic of South Africa (Fig 1): 41 asteroid, 90 crinoid, 23 echinoid, 70 holothuroid and 88 ophiuroid specimens. All specimens studied here are invertebrates and were collected legally with written permissions from the regional and national South African authorities. The permit number of the expedition of 2016 is RES 2016–02 and was granted by the Department of

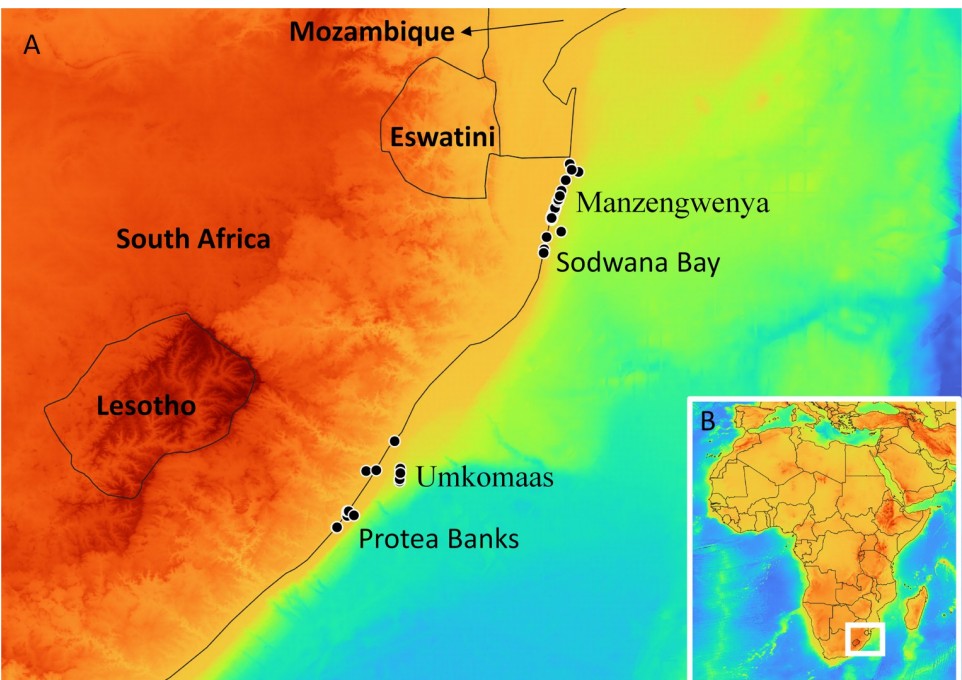

**Fig 1. Sampling locations of the specimens sequenced in this study.** A. Map of the East Coast of South Africa with sampling locations represented as black dots, country names in bold, and sampling location names in normal font. B. Map of Africa where the zoomed geographic region is represented as a white rectangle outline. This map was created using the software QGIS v. 3.22.6 [30], the GEBCO Grid [31] and Natural Earth [32].

Agriculture, Forestry and Fisheries (DAFF) and the Department of Environmental Affairs (DEA) of South Africa to Yves Samyn. The specimens were collected in the eulittoral zone whilst snorkelling or in the sublittoral zone to ca. 40 m depth while scuba diving. Specimens were anesthetized with ± 5% $Mg_2.6H_20$ (or freshwater for crinoids and some ophiuroids), fixed with 80–95% ethanol for at least 24 hours, rinsed with 70–75% ethanol and finally stored either in 70–75% ethanol or dry. Tissue samples, generally dermis, podia or tips of arms were sampled in the field and stored in absolute ethanol for subsequent DNA analysis. They were deposited in the Royal Belgian Institute of Natural Sciences (Belgium), the Royal Museum for Central Africa (Belgium) and the University of KwaZulu-Natal and the Iziko Museums of South Africa (Republic of South Africa). Voucher numbers and specimen information are provided in S1 Table. Taxonomical identifications were performed on the bases of external and internal anatomy and internal skeletal elements. For this, specimens were either denuded from their dermis to reveal the underlying calcareous skeleton (for instance test and spine structure of echinoids) or dissected to isolate various (microscopic) calcareous deposits (for instance the ossicle assemblage of holothuroids). Identifications were achieved by eye, stereo light microscope or scanning electron microscope. General identification keys [27, 28] were complemented with more detailed taxon- and region-specific revisions as well as with more recent taxonomic literature. Taxonomy is following the World Register of Marine Species [29]. When possible, specimens were compared with vouchers stored in museum collections. These identifications are hereunder designated as "morphospecies". In some cases, specimens could not be identified to the species level, either because diagnostic taxonomic characters could not be assessed or because of differences between the specimen and species descriptions. In these cases, provisional identifications were given the taxonomic rank with the highest confidence (e.g. *Ophiocoma* sp. or Amphiuridae sp.).

## DNA data collection

Total genomic DNA was extracted using the NucleoSpin Tissue Kit (Macherey-Nagel, Germany) following the standard protocol for animal tissue. Fragments of the 5' end of the cytochrome *c* oxidase subunit I (COI) gene were amplified and sequenced using the following pairs of primers (sequence provided from 5' to 3'): LCO1490 (`GGTCAACAAATCATAAAGAT ATTGG`) with HCO2198 (`TAAACTTCAGGGTGACCAAAAAATCA`) [33], EchinoF1 (`TTTCAAC TAATCATAAGGACATTGG`) with EchinoR1 (`CTTCAGGGTGTCCAAAAAATCA`) [10], COIeF (`ATAATGATAGGAGGRTTTGG`) with COIeR (`GCTCGTGTRTCTACRTCCAT`) [34] and LoboF1 (`KBTCHACAAAYCAYAARGAYATHGG`) with LoboR1 (`TAAACYTCWGGRTGWCCRAARAAYCA`) [35]. When these primer pairs were not successful, other combinations of forward and reverse primers were tested and provided amplicons (LCO1490/EchinoR1, LCO1490/COIeR, EchinoF1/HCO2198, EchinoF1/COIeR COIeF/HCO2198 and COIeF/EchinoR1). The polymerase chain reactions (PCR) were prepared in volumes of 25 μl containing 2 μl of DNA template, 0.03 U/μl of Platinum® Taq DNA Polymerase (Life Technologies, USA), 1X PCR buffer, 0.2 mM dNTPs, 0.4 (all primer pairs except LoboF1/LoboR1) or 0.6 (LoboF1/LoboR1) μM of each primer and 1.5 mM MgCl$_2$. The PCR profiles started with one step at 94˚C for 3 min. Then, for all primer pairs except LoboF1/LoboR1, the next step consisted of 40 cycles at three temperatures: 94˚C for 30 s, 48˚C for 30 s and 72˚C for 45 s. For the primer pair LoboF1/LoboR1, five first cycles were performed at 94˚C for 30 s, 45˚C for 90 s and 72˚C for 60 s, before 40 additional cycles at 94˚C for 30 s, 54˚C for 90 s and 72˚C for 60 s. All PCR profiles ended with a final step at 72˚C for 7 min. PCR products were visualized using 1.2% agarose gel electrophoresis and purified using the ExoSAP procedure (Exonuclease I—Shrimp Alkaline Phosphatase from ThermoFisher, USA). PCR products were sequenced bi-directionally on an ABI automated capillary sequencer (ABI3130xl) using the BigDye v3.1 chemistry following the manufacturer's instructions (Life Technologies, USA). DNA chromatograms were checked, trimmed and assembled using CodonCode Aligner© v8.0.2 (CodonCode Corp., Centerville, Massachusetts). Consensus sequences were inspected (detection of gaps and stop codons) and compared to public records using the Basic Local Alignment Search Tool (BLAST) [36] of the National Centre for Biotechnology Information, U.S. National Library of Medicine (NCBI) to detect and exclude obvious contaminations.

## DNA barcode analysis

The COI sequences generated here were compared with those available in the Barcode of Life Data System (BOLD) [37]. For this, the "advanced search of records" was used from the workbench, whereby the name of each class was searched in the field "taxonomy" and both the target marker "COI-5P" and the option "include public records" were selected. All retrieved records were downloaded after applying the built-in filters to exclude the sequences with stop codons and tagged as contaminants. For each class, a dataset was assembled, containing both the public sequences (below called "BOLD sequences") and those obtained here, aligned using the MUSCLE algorithm [38] on MEGA v7.0.26 [39]. Shorter sequences were removed to keep at least 150 bp common to all aligned sequences. Proportions of differences (uncorrected p-distances) among all sequences in each alignment were calculated using the package ape [40] in the R language and environment for statistical computing and graphics. The "Search Taxonomy" tool of BOLD [25] was used on the 24 May 2022 to check the availability of public COI sequences for each species identified here. For specimens identified with provisional species names, the identification engine of BOLD was used to search for best matches in the "All Barcode Records" database on BOLD, which includes COI sequences that have not yet been

released publicly ("private records"). The Process IDs of the DNA sequences in the Barcode of Life Data Systems (BOLD) are from CHARE001-20 to CHARE400-20 (project CHARE).

## COI-based species delimitation

We applied four methods to delimit putative species on the basis of DNA barcodes and without *a priori* knowledge of species identity: (i) the automatic barcode gap discovery (ABGD) method [41], (ii) the "Barcode Index Numbers" (BIN) based on Refined Single Linkage (RESL) analysis [42], (iii) the General Mixed Yule Coalescent (GMYC) model [43] and (iv) a Bayesian implementation of the Poisson tree processes (bPTP) model [44]. These delineations of putative species based on DNA barcodes represent a repeatable way to recognize operational taxonomic units (OTUs), which can be used with other lines of evidence for species assignment [45]. The four methods were applied to each class separately. The first two methods are based on distances among COI sequences and the two others are based on the phylogenetic species concept for which an inference of phylogeny was performed (see below). For the ABGD method, the web application [46] was used to enable the automatic detection of gaps in the distribution of pairwise distances among DNA barcodes, which can be used to delimit putative species. The BINs were obtained on BOLD., BINs are regarded as proxies to taxonomic species. These BINS result from a single linkage clustering of the sequences, which were then grouped or split based on their similarity and connectivity. When the sequence overlap with the BOLD sequences was lower than 300 bp and a BIN was not attributed directly by BOLD, the BIN of the best match above 99% similarity was assigned to the record. For the GMYC method, we used the R package "splits" with the single threshold option [47] to determine the transition point between inter- and intra-species branching rates on a time-calibrated ultrametric tree [43] constructed as detailed in the section below. In the bPTP model, the transition points between inter- and intra-species branching are estimated from mean expected number of substitutions per site between two branching events [44]. Congruence among the species identifications based on morphology and on COI sequences was analysed and discordances were classified in three categories: (R) "resolution issues" when representatives of the same morphospecies were subdivided in several DNA-based putative species by the different species delimitation methods; (S) "splitting issues" when these representatives were split in several DNA-based putative species by all species delimitation methods and (L) "lumping issues" when different morphospecies were grouped in one putative species by all DNA-based species identification methods. We also tagged the discordance if it was observed only within the data produced here (H) or only after merging our data with all data (A) available online.

## Phylogenetic analysis

Phylogenetic trees were reconstructed through Bayesian inference (BI) and were used as an input for both GMYC and bPTP. The program MrBayes v3.2.7a [48] was used with the strict clock model [49]. The analysis was run on the CIPRES Science Gateway [50] using the best partition scheme and best-fit substitution models estimated using PartitionFinder v. 1.1 [51]. As datasets included 1384–4898 sequences, we saved computation time by performing the BI with representative unique haplotypes collected as follows: the ABGD approach was applied to the whole dataset using parameters that tend to oversplit the samples (prior maximal p-distance (P) ranging from 0.0001 to 0.1 and a relative gap width of 1). The oversplit resulting from the partition with a P = 0.0001 was checked in the graph representing the number of groups obtained for the selected range of prior intraspecific divergences. Three sequences were then collected randomly from each of these groups (less when the group consisted in one or two sequences). Unique haplotypes were then extracted from this selection using the R

language and environment and the pegas package [52]. For BI, two parallel runs were run for 50 million generations. Convergence was checked and the first 25% of the tree sampled were discarded as "burn-in" in order to discard the preliminary steps, during which the chain moves from its unrepresentative initial value to the modal region of the posterior [53]. The majority-rule consensus trees reconstructed here are available as S1–S5 Files.

## Results

### Overall results

Based on morphology, the 312 echinoderm specimens sampled in this study were assigned to 103 morphospecies belonging to the five extant classes of Echinodermata (Tables 1 and 2). Of these 103 morphospecies, 18 could not be assigned to any described species and were given a provisional name (Tables 1 and 2). The 312 DNA sequences generated here were fragments of 420 to 841 base pairs (bp) (arithmetic mean of 692 bp) of the 5' end of the cytochrome *c* oxidase subunit I gene (COI). For the vast majority of the morphospecies (91/103), COI sequences showed at least one substitution compared with the sequences available so far in public repositories. For 46 morphospecies, COI sequences even showed more than 5% p-distance difference with their best match in public repositories (Fig 2). For 30 of the 85 morphospecies that could be assigned to a described species, we publish here the first COI sequences (Table 2). The sequences were grouped by the four species delimitation methods in 99 (ABGD), 103 (GMYC and bPTP) and 105 (BIN) putative species (Table 2).

### Asteroidea

Among the 41 starfishes sequenced here, 18 different morphospecies were identified: 15 were recognized as species based on the current taxonomy and three were identified with provisional names (*Aquilonastra* cf. *rowleyi*, *Aquilonastra* sp. and *Leiaster* cf. *leachi*). Species delimitation methods assigned the 41 COI sequences to a total of 18 (ABGD), 20 (GMYC and bPTP) or 21 (BIN) putative species (Table 1). Discordances between morphospecies assignments and DNA-based identification methods are detailed in Table 3. The number of putative species determined with GMYC, bPTP and BIN was higher than with ABGD because *Fromia milleporella* was subdivided in two putative species by all methods except ABGD, and *Gomophia egyptiaca* was lumped with *Nardoa variolata* in one putative species by ABGD only. BIN was the only method to split the sequences obtained for *Aquilonastra* sp. in two putative species. When also considering BOLD data, we note that the sequences of four species were subdivided in two or more putative species (Table 3): Our sequences of *Fromia indica*, *Ophidiaster hemprichi* and *Echinaster purpureus* were in a putative species separated from those from the Western

**Table 1. Number and identification of specimens, morphospecies and putative species estimated from the DNA sequences obtained in this study.**

| Class | Number of specimens | Number of morphospecies[1] | Number of morphospecies with a provisional identification | Number of putative species according to method ABGD/GMYC/bPTP/BIN |
|---|---|---|---|---|
| **Asteroidea** | 41 | 18 | 3 | 18/20/20/21 |
| **Crinoidea** | 90 | 8 | 1 | 8/10/10/10 |
| **Echinoidea** | 23 | 12 | 3 | 11/11/11/12 |
| **Holothuroidea** | 70 | 32 | 6 | 31/31/31/29 |
| **Ophiuroidea** | 88 | 33 | 5 | 31/31/31/33 |
| **Total** | 312 | 103 | 18 | 99/103/103/105 |

[1]: The number of morphospecies includes the number of provisional identifications

**Table 2. Taxonomic classification of the morphospecies identified in this study.**

| Class | Order | Family | Species | N. |
|---|---|---|---|---|
| Asteroidea | Spinulosida Perrier, 1884 | Echinasteridae Verrill, 1870 | *Echinaster purpureus* (Gray, 1840) | 5 |
| | Valvatida Perrier, 1884 | Asterinidae | *Aquilonastra* cf. *rowleyi* O'Loughlin & Rowe, 2004 | 4 |
| | | | *Aquilonastra* sp. | 2 |
| | | Goniasteridae Forbes, 1841 | *Fromia indica* (Perrier, 1869) | 2 |
| | | | *Fromia milleporella* (Lamarck, 1816) | 2 |
| | | | *Fromia schultzei* Döderlei, 1910 | 1* |
| | | Mithrodidae Viguier, 1878 | *Thromidia seychellesensis* Pope & Rowe, 1977 | 2* |
| | | Ophidiasteridae Verrill, 1870 | *Cistina columbiae* Gray, 1840 | 1 |
| | | | *Ferdina sadhaensis* Marsh & Campbell, 1991 | 3* |
| | | | *Gomophia egyptiaca* Gray, 1840 | 3* |
| | | | *Leiaster* cf. *leachi* (Gray, 1840) | 1 |
| | | | *Linckia laevigata* (Linnaeus, 1758) | 3 |
| | | | *Nardoa variolata* (Retzius, 1805) | 4* |
| | | | *Ophidiaster hemprichi* Müller & Troschel, 1842 | 1 |
| | | | *Ophidiaster perrieri* de Loriol, 1885 | 1* |
| | | Oreasteridae Fisher, 1911 | *Choriaster granulatus* (Lütken, 1869) | 1* |
| | | | *Monachaster sanderi* (Meissner, 1892) | 4 |
| | | | *Protoreaster lincki* (de Blainville, 1830) | 1* |
| Crinoidea | Comatulida Clark, 1908 | Antedonidae Norman, 1865 | *Annametra occidentalis* (A.H. Clark, 1915) | 5 |
| | | Colobometridae A.H. Clark, 1909 | *Cenometra emendatrix* (Bell, 1892) | 4* |
| | | | *Oligometra serripinna* (Carpenter, 1881) | 21 |
| | | Comatulidae Flemming, 1828 | *Comanthus wahlbergii* (Müller, 1843) | 15 |
| | | | *Comanthus* sp. | 1 |
| | | Mariametridae A.H. Clark, 1809 | *Dichrometra palmata* (Müller, 1841) | 12 |
| | | | *Stephanometra indica* (Smith, 1876) | 1 |
| | | Tropiometridae A.H. Clark, 1908 | *Tropiometra carinata* (Lamarck, 1816) | 31 |
| Echinoidea | Camarodonta Jackson, 1912 | Echinometridae Gray, 1855 | *Echinometra mathaei* (de Blainville, 1825) | 3 |
| | | | *Echinostrephus molaris* (de Blainville, 1825) | 4* |
| | | Temnopleuridae A. Agassiz, 1872 | *Microcyphus rousseaui* L. Agassiz, in Agassiz & Desor, 1846 | 1* |
| | | | *Temnopleurus* cf. *toreumaticus* (Leske, 1778) | 1 |
| | | Toxopneustidae Troschel, 1872 | *Tripneustes gratilla* (Linnaeus, 1758) | 4 |
| | Cidaroidea Claus,1880 | Cidaridae Gray, 1825 | *Eucidaris metularia* (de Lamarck, 1816) | 1 |
| | | | *Phyllacanthus imperialis* (de Lamarck, 1816) | 1* |
| | Diadematoida Duncan, 1889 | Diadematidae Gray, 1855 | *Diadema savignyi* (Audouin, 1829) | 4 |
| | | | *Diadema* sp. | 1 |
| | Echinoneoida H.L. Clark, 1925 or Clypeasteroida Agassiz, 1872 | Irregularia sp. | | 1 |
| | Spatangoida L. Agassiz 1840 | Eurypatagidae Kroh, 2007 | *Eurypatagus parvituberculatus* (H.L. Clark, 1924) | 1* |
| | Stomopneustoida Kroh & Smith, 2010 | Stomopneustidae Mortensen, 1903 | *Stomopneustes variolaris* (de Lamarck, 1816) | 1 |
| Holothuroidea | Aspidochirotida Grube, 1840 | Holothuriidae Ludwig, 1894 | *Actinopyga echinites* (Jaeger, 1833) | 6 |
| | | | *Actinopyga mauritiana* (Quoy & Gaimard, 1834) | 2 |
| | | | *Actinopyga obesa* (Selenka, 1867) | 3 |
| | | | *Bohadschia* sp. | 1 |
| | | | *Bohadschia subrubra* (Quoy & gaimard, 1834) | 2 |
| | | | *Holothuria albofusca* Cherbonnier, 1988 | 2* |
| | | | *Holothuria atra* Jaeger, 1833 | 8 |
| | | | *Holothuria cinerascens* (Brandt, 1835) | 1 |

*(Continued)*

**Table 2.** (Continued)

| Class | Order | Family | Species | N. |
|---|---|---|---|---|
| | | | *Holothuria difficilis* Semper, 1867 | 1 |
| | | | *Holothuria edulis* Lesson, 1830 | 1 |
| | | | *Holothuria hilla* Lesson, 1830 | 8 |
| | | | *Holothuria impatiens* Selenka, 1867 | 2 |
| | | | *Holothuria insignis* Ludwig, 1875 | 2 |
| | | | *Holothuria leucospilota* (Brandt, 1835) | 2 |
| | | | *Holothuria lineata* Ludwig, 1875 | 1 |
| | | | *Holothuria nobilis* (Selenka, 1867) | 2 |
| | | | *Holothuria pardalis* Selenka, 1867 | 7 |
| | | | *Holothuria pervicax* Selenka, 1867 | 2 |
| | | | *Holothuria rigida* (Selenka, 1867) | 1* |
| | | | *Holothuria.* sp. AB48960534 RMCA.2121 | 1 |
| | | | *Holothuria.* sp. AB49115697 RMCA.2103 | 1 |
| | | | *Holothuria.* sp. AB49115712 RMCA.2129 | 1 |
| | | | *Holothuria* sp. AB49115671 RMCA.2634 | 1 |
| | | | *Holothuria tuberculata* Thandar, 1984 | 1* |
| | | | *Labidodemas pertinax* (Ludwig, 1875) | 1 |
| | Dendrochirotida Grube, 1840 | Cucumariidae Ludwig, 1894 | *Pseudocnella sykion* (Lampert, 1885) | 1* |
| | | | *Trachasina crucifera* (Semper, 1869) | 2 |
| | | Phyllophoridae Oestergren, 1907 | *Massinium arthroprocessum* (Thandar, 1989) | 1* |
| | | | *Massinium maculosum* Samyn & Thandar, 2003 | 2* |
| | | | *Stolus buccalis* (Stimpson, 1855) | 1* |
| | | | *Thyone* sp. | 1 |
| | | Sclerodactylidae Panning, 1949 | *Ohshimella ehrenbergii* (Selenka, 1868) | 2* |
| Ophiuroidea | Amphilepidida O'Hara, Hugall, Thuy, Stöhr & Martynov, 2017 | Amphiuridae Ljungman, 1867 | *Amphilimna cribriformis* H.L. Clark, 1974 | 1* |
| | | | *Amphiuridae* sp. | 2 |
| | | Ophiactidae Matsumoto, 1915 | *Ophiactis picteti* (Müller & Troschel, 1842) | 4* |
| | | | *Ophiactis savignyi* (Müller & Troschel, 1842) | 1 |
| | | | *Ophiactis* sp. | 1 |
| | | Ophiolepididae Ljungman, 1867 | *Ophiolepis cincta* Müller & Troschel, 1842 | 5 |
| | | | *Ophioplocus imbricatus* Lyman, 1862 | 1 |
| | | Ophionereididae Ljungman, 1867 | *Ophionereis dubia* (Müller & Troschel, 1842) | 1 |
| | | | *Ophionereis porrecta* Lyman, 1861 | 8 |
| | | Ophiotrichidae Ljungman, 1867 | *Macrophiothrix demessa* (Lyman, 1862) | 1 |
| | | | *Macrophiothrix longipeda* (de Lamarck, 1816) | 2 |
| | | | *Macrophiothrix propinqua* (Lyman, 1862) | 1 |
| | | | *Ophiothela* sp. | 2 |
| | | | *Ophiothela venusta* (de Loriol, 1900) | 4* |
| | | | *Ophiothrix echinotecta* Balinsky, 1957 | 3* |
| | | | *Ophiothrix foveolata* Marktanner-Turneretscher, 1887 | 2* |
| | | | *Ophiothrix fragilis* Abildgaard in O.F. Müller, 1789 | 1 |
| | | | *Ophiothrix purpurea* von Martens, 1867 | 6 |
| | | | *Ophiothrix trilineata* Lütken, 1869 | 2 |
| | Euryalida De Lamarck, 1816 | Gorgonocephalidae Ljungman, 1867 | *Astroboa nuda* (Lyman, 1874) | 2 |
| | | | *Gorgonocephalidae* sp. | 1 |

*(Continued)*

**Table 2.** (Continued)

| Class | Order | Family | Species | N. |
|---|---|---|---|---|
| | Ophiacanthida O'Hara, Hugall, Thuy, Stöhr & Martynov, 2017 | Ophiocomidae Ljungman, 1867 | *Ophiocoma brevipes* Peters, 1851 | 2 |
| | | | *Ophiocoma cynthiae* Benavides-Serrato & O'Hara, 2008 | 2 |
| | | | *Ophiocoma doederleini* de Loriol, 1899 | 1* |
| | | | *Ophiocoma erinaceus* Müller & Troschel, 1842 | 7 |
| | | | *Ophiocoma pusilla* (Brock, 1888) | 1 |
| | | | *Ophiocoma* sp. | 2 |
| | | | *Ophiocoma valenciae* Müller & Troschel, 1842 | 1* |
| | | | *Ophiomastix koehleri* Devaney, 1977 | 2* |
| | | Ophiodermatidae Ljungman, 1867 | *Ophiarachnella gorgonia* (Müller & Troschel, 1842) | 3 |
| | | | *Ophiarachnella septemspinosa* (Müller & Troschel, 1842) | 4* |
| | | | *Ophiopeza fallax* Peters, 1851 | 9 |
| | | | *Ophiopeza spinosa* (Ljungman, 1867) | 3 |

N. Number of specimens representing each morphospecies.

*. Morphospecies for which no public COI data is available in BOLD (on 20 April 2022).

Pacific Ocean or the coasts of Saudi Arabia. For *Linckia laevigata* only a few BOLD sequences were separated from conspecifics (including ours), but they did not seem to have a different geographic origin. Sequences of *Linckia laevigata* were also lumped with all BOLD sequences of *Linckia multifora* by all methods. The three morphospecies with provisional species names represented three (ABGD, GMYC and bPTP) to four (BIN) putative species that were separate

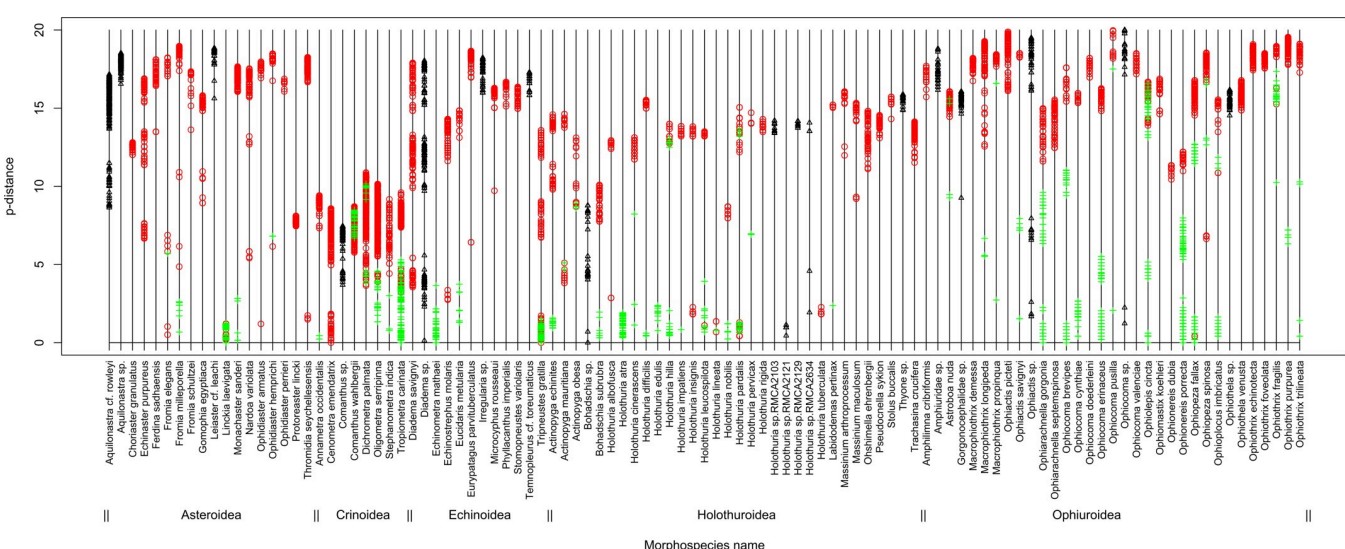

**Fig 2. Best matches between the DNA sequences generated here and those available in the Barcode of Life Data System (BOLD).** The p-distances (proportion of sites at which two sequences are different) separating each DNA sequence obtained here and its best matches in BOLD are plotted (maximum 100 best matches with a Process ID, no provisional names and a minimum of 80% similarity were considered). The p-distances are grouped for each morphospecies (x-axis) and plotted as green crosses when species names matched, as red circles when species names did not match or as black triangles when morphospecies with provisional names were included in the comparison. Distances >20% were measured between conspecifics of *Echinoneus cyclostomus*, *Holothuria difficilis*, *Holothuria impatiens* and *Ophiothrix fragilis*.

**Table 3. Discordances observed among morphospecies and DNA-based species delimitation methods.**

| Morphospecies | Description of the discordance | Categ.—Resol.–Context[1] |
|---|---|---|
| **Asteroidea** | | |
| *Echinaster purpureus* | ALL but ABGD: 5 seq. split from 1 BOLD seq. in 2 putative species | S—R—A |
| *Fromia milleporella* | ALL but ABGD: 2 seq. and 5 BOLD seq. split in 2 (GMYC and bPTP) to 3 (BIN) putative species | S—R—H |
| *Fromia indica* | ALL but ABGD: 2 seq. and 3 BOLD seq. split in 2 putative species | S—R—A |
| *Gomophia egyptiaca* and *Nardoa variolata* | ABGD: seq. of both morphospecies assigned to the same putative species, which also includes BOLD seq. identified as *Gomophia* sp. (ECLI009-08 49) and *Nardoa novaecaledoniae* (GBEH0086-06) | L—R—H |
| *Linckia laevigata* | ALL: 3 seq. and >800 BOLD seq. lumped with *Linckia multifora* in the same putative species | L—_—A |
| | ALL but ABGD: 3 seq. and 876 BOLD seq. split in >1 putative species | S—R—A |
| *Ophidiaster hemprichi* | ALL: 1 seq. split from 15 BOLD seq. and forming 2 putative species | S—_—A |
| **Crinoidea** | | |
| *Cenometra emendatrix* | ALL: 5 seq. lumped with 2 (bPTP and GMYC) or 3 (ABGD and BIN) BOLD seq. of *Cenometra bella* in 1 putative species | L—_—A |
| *Comanthus wahlbergii* | ALL: 14 seq. and 8 BOLD seq. split in 2 putative species | S—_—H |
| *Dichrometra palmata* | ALL but ABGD: 12 seq. of *D. palmata* split from 1 BOLD seq. of *D. palmata* in 2 putative species | S—R—A |
| | ALL: one additional BOLD seq. of *D. palmata* (GBEH4375-13) split in an additional putative species | S—_—A* |
| *Oligometra serripinna* | ALL but ABGD: 21 seq. and 1 BOLD seq. split in 2 putative species | S—R—H |
| *Stephanometra indica* | ALL but ABGD: 1 seq. and 2 BOLD seq. split from a third BOLD seq. in 2 putative species | S—R—A |
| *Tropiometra carinata* | ALL but ABGD: 34 seq. split in 2 putative species (3 when considering 32 BOLD seq.) | S—R—H |
| | ABGD: all seq. of *Tropiometra carinata* in 1 putative species, but lumped with *T. afra* and *T. macrodiscus* | L—R—A |
| **Echinoidea** | | |
| *Diadema savignyi* | ALL but BIN: 6 seq. of *Diadema savignyi* lumped with BOLD seq. of *D. mexicanum*, *D. antillarum* and *Echinothrix diadema* | L—R—A |
| *Echinometra mathaei* | ABGD and GMYC: 3 seq. and >100 BOLD seq. lumped with 9 seq. of *E. oblonga* | L—R—A |
| | bPTP and BIN: 3 seq. and >100 BOLD seq. split in 3 or more putative species | S—R—A |
| *Echinostrephus molaris* | ABGD and GMYC: 5 seq. lumped with 1 BOLD seq. of *Heterocentrotus trigonarius* (misidentification tag in BOLD) | L—R—A* |
| *Eucidaris metularia* | BIN: 1 seq. and 7 BOLD seq. split in 3 BINs | S—R—A |
| *Tripneustes gratilla* | All: 5 seq. and 44 BOLD seq. lumped with BOLD seq. of *T. depressus* and *T. kermadecensis* | L—_—A |
| **Holothuroidea** | | |
| *Actinopyga mauritiana* | ALL: 2 seq. split from 7 BOLD seq, forming 2 (ABGD) or 3 (ALL but ABGD) putative species | S—_—A |
| *Actinopyga obesa* | ALL: 2 seq. split from 1 BOLD seq, forming 2 putative species | S—_—A |
| *Holothuria albofusca* and *H. hilla* | ALL: 2 seq. of *H. albofusca* lumped with 3 seq. (1 BOLD) of *H. hilla* | L—_—H |
| *Holothuria atra* | ALL: 8 seq. and 109 BOLD seq. split in 2–3 putative species (only an additional species within BOLD records) | S—_—A |
| *Holothuria cinerascens* | ALL: 1 seq. split from the other BOLD seq. | S—_—A |
| *Holothuria difficilis* | ALL: 2 seq. split from 1 BOLD seq. forming 2 putative species | S—_—A* |
| *Holothuria edulis* | BIN: 1 seq. and 23 BOLD seq. split in 2 putative species (only a second species within BOLD records) | S—R—A |
| | ALL but BIN: one additional divergent BOLD seq. forming a third putative species | |
| *Holothuria hilla* | ALL: 9 seq. and 5 BOLD seq. split in 4 putative species | S—_—H |
| *Holothuria hilla, H. impatiens, H. insignis* and *H. pardalis* | ALL: 1 seq. of *H. hilla*, 1 seq. of *H. pardalis*, 1 seq. of *H. impatiens* and 1 seq. of *H. insignis* lumped in 1 putative species | L—_—H |
| *Holothuria impatiens* | ALL: 2 seq. and 8 BOLD seq. split in 5 (ABGD, GMYC and BIN) or 6 (bPTP) putative species | S—_—H |
| *Holothuria insignis* | ALL: 2 seq. split in two putative species | S—_—H |
| *Holothuria insignis, H. lineata, H. pardalis, H. tuberculata* | ALL: 3 seq. of *H. pardalis*, 1 of *H. insignis*, 2 of *H. lineata* and 1 of *H. tuberculata* lumped in 1 putative species | L—_—H |
| *Holothuria nobilis* | bPTP: 2 seq. and 10 BOLD seq. split in 2 putative species | S—R—A |
| *Holothuria pardalis* | ALL: 7 seq. and 1 BOLD seq. split in 4 putative species | S—_—H |

*(Continued)*

**Table 3.** (Continued)

| Morphospecies | Description of the discordance | Categ.—Resol.–Context[1] |
|---|---|---|
| **Ophiuroidea** | | |
| *Astroboa nuda* | ALL: 3 seq. split from 3 BOLD seq. forming a total of 3 separate putative species | S—_—A |
| | ALL: one additional divergent BOLD seq. (ECHOZ007-09) forming a third putative species | S—_—A* |
| *Macrophiothrix longipeda* | ALL: 2 seq. and 2 BOLD seq. split in 3 putative species | S—_—A |
| *Macrophiothrix propinqua* | ALL: 1 seq. and 3 BOLD seq. split in 2 putative species | S—_—A |
| *Ophiactis picteti* | BIN: 5 seq. split in 3 putative species | S—R—H |
| *Ophiactis savignyi* | ALL: 1 seq. and 9 BOLD seq. split in 2 or more putative species | S—_—A |
| *Ophiarachnella gorgonia* | ALL: 3 seq. and 31 BOLD seq. split in 4 (ABGD & GMYC) or 5 (bPTP & BIN) putative species | S—_—A |
| *Ophiocoma pusilla* | ALL: 1 seq. and 1 BOLD seq. of *O. pusilla* split from another BOLD seq. of *O. pusilla* | S—_—A |
| *Ophiocoma brevipes* | ALL: 3 seq. and 42 BOLD seq. split in 2 putative species | S—_—A |
| *Ophiocoma erinaceus* | ALL but ABGD: 7 seq. and 31 BOLD seq. split in 2 putative species | S—R—A |
| *Ophiolepis cincta* | ALL: 5 seq. and 24 BOLD seq. split in 7 putative species | S—_—A |
| *Ophionereis porrecta* | ALL: 8 seq. and 42 BOLD seq. split in 9 to 10 putative species | S—_—A |
| *Ophiopeza fallax* | ALL: 9 seq. and 26 BOLD seq. split in 2 putative species | S—_—A |
| *Ophiopeza spinosa* | ALL: 3 seq. split from 2 BOLD seq. forming a total of 3 separate putative species | S—_—A |
| *Ophioplocus imbricatus* | ALL: 1 seq. and 31 BOLD seq. split in 2 putative species | S—_—A |
| *Ophiothrix fragilis* | ALL: 1 seq. split from 42 BOLD seq. and forming a total of 4 to 5 putative species | S—_—A |
| *Ophiothrix purpurea* | ALL: 6 seq. split from 3 BOLD seq. in 2 putative species | S—_—A* |
| | ALL: 1 additional BOLD seq. ECHOZ070-09 representing 1 additional putative species | |
| *Ophiothrix trilineata* | ALL: 2 seq. and 2 BOLD seq. split in 2 putative species | S—_—A |

[1] Discordances among species identifications obtained using morphology (morphospecies) and species delimitation methods based on the 5' end of the cytochrome c oxidase subunit I (COI) gene for data obtained here and retrieved from BOLD. Categ. Discordance category (S for splitting or L for lumping issue). Resol. Issue related to a different resolution of the different COI-based identification methods (R) or not (_). Context. Discordance observed within the sequences obtained here (H) or among all sequences (A).

*. Probable misidentification. "seq.". Sequence obtained here. "BOLD seq.". Sequence retrieved from BOLD.

from any other species for which COI sequences were available (Table 4). Indeed, *Aquilonastra* sp. represented one or two separate putative species (see above). *Leiaster* cf. *leachi* also formed a separate putative species, with *Leiaster glaber* Peters, 1852 (distance of 10.34%) as the closest published record. We noted a closer match (1.08–1.68%) with two private BOLD sequences labelled as *L. leachi* using the identification engine of BOLD. Finally, the specimens identified as *Aquilonastra* cf. *rowleyi* showed COI sequences that were split in two putative species, one of which was separate from any other putative species, the other was grouped with BOLD records also labelled with a provisional name (*Aquilonastra* sp.2).

## Crinoidea

The 90 crinoids analysed here were assigned to eight morphospecies, all of them corresponding to described species except one, *Comanthus* sp. Our crinoid specimens were grouped into eight (ABGD) or ten (GMYC, bPTP and BIN) putative species by the species delimitation methods (Table 1) because *Oligometra serripinna* and *Tropiometra carinata* were both subdivided in two putative species by the latter three methods. The specimens of *Oligometra serripinna* sequenced here and split in two putative species were both found in the iSimangaliso Wetland Park, not far from Sodwana Bay (Fig 1A). One of these putative species also included BOLD records from Singapore. For *Tropiometra carinata*, specimens of one putative species

**Table 4. Specimens for which a provisional morphospecies name was assigned.**

| Specimen ID | Sample ID | Morphospecies | DNA-based species delimitation | BOLD identification engine (dissimilarity) |
|---|---|---|---|---|
| **Asteroidea** | | | | |
| RSAKZN/2016.026 | AB42608987 | *Aquilonastra* cf. *rowleyi* | Separate putative species including BOLD record with provisional name "*Aquilonastra* sp.2" | No better match |
| RSAKZN/2016.025 | AB42608993 | *Aquilonastra* cf. *rowleyi* | | *Asterina conandae* (2.71–2.86) |
| RSAKZN/2016.087 | AB42609757 | *Aquilonastra* cf. *rowleyi* | | |
| RSAKZN/2016.027 | AB42608981 | *Aquilonastra* cf. *rowleyi* | Separate putative species | |
| RSAKZN/2016.146 | AB42609998 | *Aquilonastra* sp. | 1 or 2 (BIN) separate putative species | No match |
| RSAKZN/2016.189 | AB42610004 | *Aquilonastra* sp. | | |
| RSAKZN/2016.136 | AB42609907 | *Leiaster* cf. *leachi* | Separate putative species | *Leiaster leachi* (1.08–1.68) |
| **Crinoidea** | | | | |
| RSAKZN/2000.081 | AB49142178 | *Comanthus* sp. | Separate putative species (except for ABGD: *Comanthus suavia* and *C. parvicirrus*) | Same two species (3.69–4.59) |
| **Echinoidea** | | | | |
| RSAKZN/2016.067 | AB42609731 | *Diadema* sp. | *Echinothrix diadema* | No better match |
| RSAKZN/2016.042 | AB42608985 | *Irregularia* sp. | Separate putative species | No match |
| RSAKZN/2016.485 | AB42610002 | *Temnopleurus* cf. *toreumaticus* | Separate putative species | No match |
| **Holothuroidea** | | | | |
| RSAKZN/1999 (RMCA.2093) | AB49115693 | *Bohadschia* sp. | *Bohadschia cousteaui* | No better match |
| RSAKZN/1999 (RMCA.2121) | AB48960534 | *Holothuria* sp. RMCA.2121 | *H. pardalis*, *H. insignis*, *H. lineata* and *H. tuberculate* | No better match |
| RSAKZN/1999 (RMCA.2103) | AB49115697 | *Holothuria* sp. RMCA.2103 | Separate putative species | *Holothuria rigida* (0.77) |
| RSAKZN/2000 (RMCA.2129) | AB49115712 | *Holothuria* sp. RMCA.2129 | Separate putative species | *Holothuria hartmeyeri* (0.5) |
| RSAKZN/2003 (RMCA.2634) | AB49115671 | *Holothuria* sp. RMCA.2634 | *Holothuria isuga* | *Holothuria arenacava* (0.24–0.77) |
| RSAKZN/2016.165 | AB42610223 | *Thyone* sp. | Separate putative species | No match |
| **Ophiuroidea** | | | | |
| RSAKZN/2016.122 | AB42609870 | *Amphiuridae* sp. | *Ophiocoma* cf. *doederleini* | No better match |
| RSAKZN/2016.181 | AB42609983 | *Amphiuridae* sp. | *Ophiocoma* cf. *doederleini* | No better match |
| RSAKZN/2016.132 | AB42609883 | *Gorgonocephalidae* sp. | *Astroboa nuda* | No better match |
| RSAKZN/2016.128 | AB42609906 | *Ophiactis* sp. | Separate putative species | *Amphipholis squamata* (1.68 ->5) |
| RSAKZN/2016.019 | AB42608988 | *Ophiocoma* sp. | *Ophiocoma pusilla* | No better match |
| RSAKZN/2016.104 | AB42609903 | *Ophiocoma* sp. | *Ophiocoma pusilla* | No better match |
| RSAKZN/2016.190 | AB42610232 | *Ophiothela* sp. | *Ophiothela venusta* and *Ophiothrix purpurea* | No better match |
| RSAKZN/2016.206b | AB42609508 | *Ophiothela* sp. | *Ophiothela venusta* and *Ophiothrix purpurea* | No better match |

Specimen field ID (Specimen ID) and tissue sample ID (Sample ID) are given. The results of the DNA-based species delimitation methods are based on all four methods (ABGD, bPTP, GMYC and BIN) and the 5' end of the cytochrome *c* oxidase subunit I gene. The BOLD identification engine is based on the "All barcode records" database (which includes sequences that have not yet been released publicly on the 10[th] of March 2022). Species names are given when the DNA sequence queried was grouped in the same putative species as another record (by the four DNA-based species delimitation methods unless stated otherwise) or matched a described species with a dissimilarity <5% in the BOLD identification engine. "Separate putative species" means that the sequence was not grouped with another record. "No better match" means that the most similar sequences in the BOLD identification engine are identical to the results of the DNA-based species delimitation. "No match" means that no sequence was found with a dissimilarity <5%.

were collected in Protea Banks while specimens from the other putative species were from Umkomaas, Manzengwenya and Sodwana Bay (Fig 1). Compared to the data available in BOLD, discordances were noted for four further species (Table 3). Indeed, our COI sequences of *Cenometra emendatrix* were lumped with BOLD records of *C. bella*. For *Comanthus wahlbergii*, all methods split our sequences from the other sequences available online for the same species. For *Dichrometra palmata*, the sequences obtained from South Africa were split from one BOLD sequence originating from the East coast of Australia by GMYC, bPTP and BIN. Finally, the same three methods also split *Stephanometra indica* in two putative species, one of which included our sequence and three BOLD sequences from Australia and Papua New Guinea. The other was represented by one single sequence from the West Indian Ocean. The sequence obtained from the single specimens with a provisional name (*Comanthus* sp.) was either separated from all other putative species (ABGD, bPTP and GMYC) or grouped with *Comanthus suavia* and *C. parvicirrus* (ABGD), with no better match found by the BOLD identification engine among private data (Table 4).

## Echinoidea

The 23 echinoids studied here represented 12 morphospecies, three of which were given provisional identifications: *Diadema* sp., Irregularia sp. and *Temnopleurus* cf. *toreumaticus*. COI-based species delimitation methods identified 11 (ABGD, GMYC and bPTP) or 12 (BIN) putative species as BIN was the only method that did not lump *Diadema savignyi* with *Diadema* sp. (Table 3). Concerning the discordances with BOLD records, the same sequences of *Diadema savignyi* and *Diadema* sp. were also lumped by ABGD, GMYC and bPTP with BOLD sequences of *D. mexicanum*, *D. antillarum* and *Echinothrix diadema*. Similarly, sequences of *Echinometra mathaei* were either lumped with *Echinometra oblonga* (ABGD and GMYC) or split in minimum three putative species (bPTP and BIN). Our COI sequences of *Echinostrephus molaris* were lumped by ABGD and GMYC in the same putative species as one BOLD record identified as *Heterocentrotus trigonarius*, but which was tagged as misidentified. BIN was the only method to subdivide *Eucidaris metularia* in three putative species, with one putative species only represented by specimens from Hawaii, and two putative species including both specimens from Saudi Arabia (Red Sea) and South Africa. *Tripneustes gratilla* was grouped by all methods with *Tripneustes depressus* and a few sequences of *Tripneustes kermadecensis*. The COI sequences with a provisional identification (other than *Diadema* sp. already detailed above), Irregularia sp. and *Temnopleurus* cf. *toreumaticus*, were not grouped with any other COI sequence and did not match any private sequence in the identification engine of BOLD (Table 4).

## Holothuroidea

Among the 70 sea cucumbers included in this study, 32 morphospecies were identified. Six of them were attributed provisional names (*Bohadschia* sp., 4 *Holothuria* sp. and *Thyone* sp.). Species delimitation methods distinguished 29 (BIN) or 31 (ABGD, GMYC and bPTP) putative species. The discordances between the species identified morphologically and using COI were caused by ten sequences of eight morphospecies that were mixed in three putative species by all methods. A first putative species grouped *Holothuria albofusca* with *H. hilla*; a second putative species grouped *H. insignis*, *H. lineata*, *H. pardalis*, *H. tuberculata* and *H.* sp. (voucher RMCA.2121), and the third putative species grouped *H. hilla*, *H. impatiens*, *H. insignis* and *H. pardalis* (Table 3). Conversely, several of these morphospecies were split in highly divergent putative species by all methods. Large distances were found within our sampling for *Holothuria hilla* (p-distance up to 19.74%), *H. impatiens* (16.45%), *H. insignis* (13.16%) and *H.*

*pardalis* (15.79%). Large distances were also recorded for two of these species within the public data set available in BOLD (*Holothuria hilla* and *H. impatiens* with p-distances of 11.84–16.45% and 17.76–24.34%, respectively). Finally, when considering both BOLD and our dataset, additional species were split in distantly related putative species: *Holothuria difficilis* (up to 22.37%), *Actinopyga mauritiana* (15.79%), *Holothuria atra* (11.18%), *Actinopyga obesa* (7.89%) and *Holothuria cinerascens* (5.92%). One of the six specimens with a provisional identification (*Thyone* sp.), represented a putative species that was separate from any other putative species identified in BOLD or within our dataset, and had no match using the identification tool of BOLD. The sequence of two other specimens labelled as *Holothuria* sp. (RMCA.2129 and RMCA.2103) did not group with any other putative species but matched two private records of BOLD of *Holothuria hartmeyeri* and *Holothuria rigida* with dissimilarities of 0.50 and 0.77%, respectively. The three other specimens with a provisional identification were grouped with other morphospecies by all species delimitation methods: *Bohadschia* sp. was grouped with public sequences of *B. cousteaui* (BOLD Process IDs GBEHH226-13 and GBEHH227-13), *Holothuria* sp. RMCA.2634 was grouped with public sequences of *H. isuga* (BOLD Process ID GBEHH114-10) and *Holothuria* sp. RMCA.2121 was grouped with the putative species including different morphospecies mentioned above (*Holothuria insignis*, *H. lineata*, *H. pardalis* and *H. tuberculata*). This cluster also included two public records from unidentified holothuroids from Queensland, Australia (BOLD Process IDs ECLI026-08 and ECLI027-08). Using the identification tool of BOLD, two records of *Holothuria arenacava* showed a dissimilarity of 0.24–0.77% with *Holothuria* sp. RMCA.2634 (Table 4).

## Ophiuroidea

Among the 88 ophiuroids sampled here, a total of 33 morphospecies were identified. Five of these morphospecies were identified with a provisional name: Amphiuridae sp., Gorgonocephalidae sp., *Ophiactis* sp., *Ophiocoma* sp. and *Ophiothela* sp. Species delimitation methods distinguished 31 (ABGD, GMYC and bPTP) or 33 (BIN) putative species. This estimation difference is due to five COI sequences of *Ophiactis picteti* clustering in one single putative species according to ABGD, GMYC and bPTP and in three putative species according to BIN (Table 3). Remarkably, 15 ophiuroid morphospecies represented both in our sampling and in BOLD are split by all species delimitation methods in two or more putative species: *Astroboa nuda* (separated by p-distances up to 18.72%), *Macrophiothrix longipeda* (6.67%), *Macrophiothrix propinqua* (16.67%), *Ophiactis savignyi* (7.91%), *Ophiarachnella gorgonia* (10.40%), *Ophiocoma pusilla* (17.51%), *Ophiocoma brevipes* (10.98%), *Ophiolepis cincta* (16.56%), *Ophionereis porrecta* (18.11%), *Ophiopeza fallax* (12.69%), *Ophiopeza spinosa* (16.97%), *Ophioplocus imbricatus* (12.27%), *Ophiothrix fragilis* (23.85%), *Ophiothrix purpurea* (8.13% to 23.24%) if we consider a specimen that might be misidentified, see below) and *Ophiothrix trilineata* (10.37%). One additional morphospecies, *Ophiocoma erinaceus* (5.52%) was also split by GMYC, bPTP and BIN. Concerning the specimens with provisional identifications, the specimen labelled as *Ophiactis* sp. formed a putative species that was separate from all other putative species. The closest match in the BOLD identification engine was *Amphipholis squamata* with distances of 1.68–7.95. Of the two specimens identified as *Ophiothela* sp., one was grouped with four representatives of *Ophiothela venusta* and the other was grouped with six representatives of *Ophiothrix purpurea*. The specimens labelled as Amphiuridae sp.1, Gorgonocephalidae sp. and *Ophiocoma* sp. were grouped by all delimitation methods with *Ophiocoma* cf. *doederleini* (BOLD record), *Astroboa nuda* (sequenced here), and *Ophiocoma pusilla* (sequenced here), respectively (Table 4).

## Discussion

### Further taxonomic investigation required for 48 morphospecies

DNA barcoding studies represent a complementary approach to enlarge our knowledge of species biodiversity. Associating short DNA sequences to morphospecies improves the characterization of evolutionary lineages and enables the detection of both erroneous identifications, synonyms and potentially undescribed species when discordances are observed between morphospecies and COI-based putative species identifications [45]. Since COI alone is not suitable for delimiting and describing new species, the discordance observed require additional studies to redefine species on the bases of rigorous taxon delimitation methods and publicly available data [54, 55]. The total number of species identified based on morphology was overall consistent with DNA-based species delimitations (103 morphospecies versus 99–105 species estimated using COI). However, our results categorised 49 discordances (cf. "resolution", "splitting" and "lumping" issues in Table 3 and discussed below) that can be explained by differences among COI-based species identification methods, imperfect species delineation (and misidentifications) and overlooked species diversity (or species complexes). Also, the morphospecies not matching described species can either represent unknown species morphs or undescribed species. These alternative interpretations are discussed below. In total, we identified at least 48 morphospecies (including eight with a provisional name) that could benefit from additional integrative taxonomical investigations to accurately identify them to the species level.

### Discordances among COI-based species delimitation methods

All four DNA-based species delimitation methods provided congruent results for 81 of the 103 morphospecies. Part of the methods lumped four pairs of morphospecies in four putative species and split 14 morphospecies (cf. "R" issues in Table 3). Tree-based methods (GMYC, and bPTP) tend to overestimate the number of putative species while ABGD tends to underestimate them [56]. For example, *Ophiocoma erinaceus* was split in two putative species by all methods except ABGD. Recent phylogenomic data based on exon capture suggested that the two putative species found in the West Indian and in the East Indian/Pacific Oceans underwent allopatric speciation [57]. In the case of *Fromia indica*, the two different putative species identified by GMYC, bPTP and BIN were collected in different areas (South-western Indian Ocean and the Coral Sea in the South Pacific) and further investigation is necessary to know if they represent different species or divergent populations. For *O. serripinna* and *T. carinata*, also split in several putative species by all methods but ABGD, COI distances were not consistent with geographic distances, as separate putative species were collected in the same regions. A previous study including *T. carinata* from the Atlantic and Indian Oceans already revealed two divergent sympatric COI lineages, which likely represent different species [58]. Only one of these two lineages (lineage 1) included our COI sequences (from Umkomaas, Manzengwenya and Sodwana Bay). The COI sequences obtained from the Protea Banks were forming a third putative species according to GMYC, bPTP and BIN and represents another case to be further investigated. For *S. indica*, the second putative species was represented by a single BOLD sequence (GBEH3087-10) that contained four gaps, and which should be double-checked.

BIN was shown to count more putative species than GMYC and bPTP [59]. In our results, BIN was the only method in which the following morphospecies were split in two or more putative species. Sequences of *Eucidaris metularia* from the Red Sea and the Indian Ocean were split from those from the Pacific Ocean. Sequences of *Holothuria edulis* from the Red Sea

and South Africa were split from others from the Gulf of Aqaba and the more eastern part of the Indo-Pacific Ocean. The five sequences of *Ophiactis picteti* sequenced here and originating from the same locality were split in three putative species. These last seven cases, but especially the *O. picteti* case, need to be further investigated. BIN was also the only method to distinguish *Diadema savignyi*, *D. mexicanum* and *D. antillarum*, three species that hybridize and that are known to be very closely related [60, 61]. BIN and bPTP were the only methods to split *Echinometra mathaei* in three or more putative species. This is in accordance with the distinct species identified within *Echinometra mathaei* using ecological distribution, test morphology, gonadal spicules, gametes and cross-fertilization experiments [62–66]. These cases show the difficulty to interpret the different species delimitation methods in absence of additional lines of evidence and when distances among species are relatively small. Putative species are considered more reliable when supported by various delimitation methods than when supported by only one [56]. Therefore, the following discussion will focus only on putative species supported by all methods.

### Imperfect species delineation

A total of 10 morphospecies (one asteroid, one crinoid, one echinoid and seven holothuroids, Table 2) showed COI sequences that were lumped with other morphospecies. Some were also split in different putative species. Many of these cases have already been reported in the literature. In asteroids, *Linckia laevigata* and *Linckia multifora* have been extensively sequenced, with hundreds of COI sequences and a geographically comprehensive sampling in the Indo-Pacific. Both species, although morphologically recognizable based on the skeletal armature [67], are hardly distinguishable using COI sequences [19], and may both comprise several cryptic species [18, 19, 68]. Concerning the crinoid morphospecies *Cenometra emendatrix* (known from the eastern Indian Ocean), our sequences were lumped with the BOLD sequences of *Cenometra bella* (known from the central to western Indo-Pacific) in one putative species. COI sequences of the two species were separated by small p-distances (0.92–1.72%), but one COI sequence of *C. bella* from Papua New Guinea (GBMND44777-21, voucher MNHN-342) was identical to one *C. emendatrix* from South Africa. Further analysis will decipher if *C. emendatrix* and *C. bella* result from a recent allopatric speciation or represent isolated populations from one single species. In echinoids, the lumping of *Tripneustes gratilla* (from the Indian and Pacific oceans) with *Tripneustes depressus* (from the eastern Pacific) in one putative species is consistent with a previous study based on morphology, COI and bindin gene data [69], which suggested that they belong to the same species. In holothuroids, the seven problematic morphospecies identified in this study are known to be difficult to distinguish using morphological characters. For example, *Holothuria insignis* and *H. lineata* have been confirmed as valid species based upon detailed studies of the recovered type material and one newly collected voucher specimen [70, 71]. These are difficult identifications, and species belonging to the same subgenera were often mixed in some putative species (*Holothuria insignis*, *H. lineata*, *H. pardalis* and *H. tuberculata* within the subgenus *Lessonothuria* Deichmann, 1958 or *H. hilla* and *H. albofusca* within the subgenus *Mertensiothuria* Deichmann, 1958). More importantly, several of these species were both split and mixed in several divergent COI-based putative species (likely representing a series of species complexes that cannot yet be properly delineated through morphological examination (in particular the microscopic ossicle assemblage) or that were erroneously identified. Specific focus on these species already revealed that one of the COI lineages represented by *H. hilla* has been recently described as a new species (*Holothuria viridiaurantia* sp. nov.) [72]. Also, based on a combination of mitochondrial and nuclear markers, Michonneau [73] concluded that *H. impatiens* was a species

complex comprising at least 13 species. Additional investigation including further comparisons with type specimens will be necessary to reliably associate at least nine of the 10 morphospecies discussed here based on DNA data, improve species delineation and resolve sometimes extensive subjective synonymy.

## Overlooked species diversity

Most discordances between morphospecies and COI-based species delimitations were splitting issues (Table 2). Indeed, the COI sequences of 22 morphospecies (one asteroid, two crinoids, four holothuroids and 15 ophiuroids) were split in two or more putative species (Table 3) showing considerable divergences (5.92–23.85%) and deserving additional scrutiny. These cases can include both undescribed species and species already described but unrecognized morphologically and for which COI sequences are not yet available or in which divergent COI sequences coexist. For the holothuroid species *Actinopyga mauritiana*, our sequences and others from Egypt (Gulf of Aqaba) labelled as *Actinopyga* sp. nov. MA-2010 [74] showed the highest similarity with *A. varians* (4.60%) from Hawaii and Micronesia, which is considered as its sister-species based on morphology [75]. The other BOLD sequences from the Red Sea, also labelled as *A. mauritiana* could correspond to another species. Concerning *A. obesa*, the BOLD sequence from Hawaii that is separated from our sequences is grouped with *A. caerulea*, a species that was recently recognized and described [76]. It is possible that *A. caerulea* was not yet known by the identifier who assigned another species name to the BOLD sequence from Hawaii. For *Holothuria cinerascens*, the separation of one sequence from India from those from South-Africa, Pakistan, Hawaii and China suggests that *Holothuria cinerascens* is largely distributed in the Indo-Pacific and that a distinct species was collected in India. Many of the 15 ophiuroid morphospecies showing a splitting issue were already reported to be composed of several putative species [77–79]. Ophiuroid species diversity is high in the Indo-Pacific [80] and was estimated to be underestimated by 20% in the South-western Indian Ocean [77]. It is striking that the COI sequences obtained here for four ophiuroid species represented yet other putative species that were not yet sequenced. Our sequences of *Astroboa nuda*, *Ophiopeza spinosa* and *Ophiothrix purpurea* from South Africa were split from those from Australia and Japan. The sequence of *Ophiothrix fragilis* from South Africa was separated from the other *O. fragilis* sequences originating from the Atlantic and the Mediterranean Sea and forming three other COI-based putative species in our analysis. These three putative species were characterized by different geographic and bathymetric distributions [79, 81]. This underestimated species diversity in Ophiuroidea can explain why intraspecific COI distances measured for the currently described ophiuroid species were in average higher than for the other classes of echinoderms [82].

The varying levels of DNA sequence divergences observed within species may depend on sampling (representativity and geographic scale), life history and evolutionary history of the species. Species with lecithotrophic larvae such as *Ophiarachnella gorgonia*, *Ophiolepis cincta*, *Ophionereis porrecta* or *Ophiopeza fallax* are suspected to show a more limited dispersal capacity than species with planktotrophic larvae because they do not feed, and their development time is limited by their energy reserves. Therefore, species with lecithotrophic larvae often display COI with relatively larger intraspecific distances than species with planktotrophic larvae, enhancing the likeliness of allopatric speciation [9, 77, 78]. Based on these theoretical considerations, the existence of undescribed species was suspected when deep divergences were found within morphospecies with planktotrophic larvae [9]. Our results show that the South-African representatives of the four above-mentioned species with lecithotrophic larvae were indeed grouped in the same putative species as the specimens sequenced from the South-western

Indian Ocean but separated from the specimens sequenced from Australia or the Pacific Ocean. Yet, this pattern was also observed for other species with planktotrophic larvae (*Macrophiothrix longipeda*, *Macrophiothrix propinqua*, *Ophiocoma brevipes*, *Ophiocoma pusilla*, *Ophioplocus imbricatus* and *Ophiothrix trilineata*). These widespread species were suspected to be complexes of allopatrically-differentiated regional endemics by Boissin et al. [77], who also noticed morphological differences among the COI lineages of *Ophiarachnella* cf. *gorgonia*, *Ophionereis porrecta* and *Ophiolepis cincta*. However, a more comprehensive sampling might unveil haplotypes and reveal that the currently observed splits are artefacts resulting from limited sampling. Similar misleading observations were corrected for *Linckia laevigata* after a more comprehensive sampling [83]. In addition to geographic clustering, *Ophiactis savignyi*, *Ophionereis porrecta* and *Ophiolepis cincta* are represented by specimens living in the same region but showing COI sequences that were assigned to two to three different putative species (cases of putative sympatry in the Red Sea and Hawaii for the first species, Lizard Island, Australia for the second, and Nosy-Be, Madagascar, for the third one) [77]. Also, due to the evolutionary history of some species, divergent mitochondrial lineages can coexist within a species. This was shown for *Ophiura sarsii* Lütken, 1855 from the Barent Sea where distances up to 3.11% have been reported among COI sequences [84]. Therefore, the putative species determined here based on COI need to be evaluated with integrative taxonomy, taking into account the life cycle, ecology, and if necessary, additional DNA data in order to infer the evolutionary history of the species [23, 85].

In a few cases, we were able to flag possible misidentifications in BOLD. These should be verified before considering the possibility of overlooked species diversity. Sequences of *Dichrometra palmata* (GBEH4375-13), *Astroboa nuda* (ECHOZ007-09) and *Holothuria difficilis* (GBMIN138025-18) were found in putative species that are not only separated from the conspecifics sequenced here and available in BOLD, but also grouped with species that are well differentiated based on both morphology and DNA data (*Stephanometra indica* for the first one, *Holothuria pardalis* and *Astroglymma sculptum*, *Ophiocrene aenigma* and *Astroboa globifera* for the second one). Our results also support the indication in BOLD that one specimen labeled as *Heterocentrotus trigonarius* (ECLI011-08) was possibly misidentified. Other misidentifications or errors are also possible, especially for singletons or when COI sequences of related species have not yet been sequenced (e.g. *Holothuria atra* BCUH041-09, GBEH8913-19, *H. edulis* GBEH8922-19 and *Ophiothrix purpurea* ECHOZ070-09).

## Morphospecies not matching described species

Among the 18 morphospecies with a provisional name, 12 were grouped with species already described (either in the same putative species, or using the BOLD identification engine, cf. Table 4) and may thus represent yet uncharacterized phenotypes (either adults or juveniles) belonging to these described species. In the case of *Comanthus* sp., however, the distances to the closest species, *Comanthus suavia* and *Comanthus parvicirrus* (1.76–4.12%) overlap the range of distances between these two species (2.94–6.47%). Our specimen may represent a separate species of *Comanthus* because most current nominal species of this genus are widely distributed, include numerous synonyms and forms that are difficult to assign to phenotypic or interspecific variation [86]. The COI sequences of seven morphospecies (including *Aquilonastra* cf. *rowleyi* that is split in two putative species) were separated from all BOLD sequences with a species identification. These morphospecies may correspond to species that have already been described but for which COI sequences are not available. Indeed, 16% of all known echinoderm species have not been barcoded yet according to the barcode of life initiative [25]. Else, they may correspond to species that have not yet been described, as suggested

by the close match with morphospecies sequences also labelled with a provisional name by independent teams (e.g. *Aquilonastra* sp.2 in Table 4). One juvenile specimen provisionally named as Amphiuridae sp. was grouped with a BOLD record identified as *Ophiocoma* cf. *doederleini*, which belongs to the Ophiocomidae. It is possible that juveniles of Amphiuridae and Ophiocomidae share some morphological characters that become family-specific only at the adult stage.

## Conclusion

By using a simple DNA barcoding approach, our study contributed to improve our knowledge on the echinoderm biodiversity found along the East Coast of South Africa. This standard method supported by the BOLD system enables the analysis of many samples collected by different research teams at broad biogeographic scales. It confirmed, and sometimes revealed, species that should be investigated further in order to improve their delineation and describe a diversity that was not noticed so far. This is of primary importance because echinoderms represent a considerable biomass in marine habitats, play a major role in the marine ecosystems and have an economic value in the food and medical industries. The numerous putative species defined here based on COI sequences need to be better characterized and integrated in taxonomy. For this, sequencing mitochondrial and nuclear DNA from type material would provide valuable information to assign formal species names to these COI-based putative species. Keeping a sound connection between taxonomy and DNA data is of primary importance if future DNA-based investigations are expected to rely on the wide range of observations that are continuously reported for formally described species.

## Supporting information

**S1 Table. Specimen information of the 312 echinoderms specimens analysed in this study.** Columns ABGD, GMYC, bPTP and BIN provide the putative species IDs obtained with each method. Haplotype IDs are given for the representative sequences included in the phylogenetic tree reconstruction.
(XLSX)

**S1 File. Majority-rule consensus tree generated using a Bayesian inference and based on the COI dataset of Asteroidea.** The tree is in NEXUS format. Values at nodes represent posterior probabilities and tip labels include haplotype IDs, species name, origin of the data (previously in BOLD or generated in this study ("NEW") and potential species IDs according to the species delimitation methods ABGD, GMYC, bPTP and BIN. Multiple values present at the same tip are provided and separated by a '-' (for example "BOLD-NEW" means that the haplotype is represented by sequences both recovered from BOLD and sequenced here).
(TRE)

**S2 File. Majority-rule consensus tree generated using a Bayesian inference and based on the COI dataset of Crinoidea.** The tree is in NEXUS format. Nodes and tip labels are annotated as in S1 File.
(TRE)

**S3 File. Majority-rule consensus tree generated using a Bayesian inference and based on the COI dataset of Echinoidea.** The tree is in NEXUS format. Nodes and tip labels are annotated as in S1 File.
(TRE)

**S4 File. Majority-rule consensus tree generated using a Bayesian inference and based on the COI dataset of Holothuroidea.** The tree is in NEXUS format. Nodes and tip labels are annotated as in S1 File.
(TRE)

**S5 File. Majority-rule consensus tree generated using a Bayesian inference and based on the COI dataset of Ophiuroidea.** The tree is in NEXUS format. Nodes and tip labels are annotated as in S1 File.
(TRE)

## Acknowledgments

We thank Prof. Ahmed S. Thandar (School of Life Sciences, Biology, University of KwaZulu-Natal, South Africa) and Dr. Jennifer Olbers, (WILDTRUST, Durban, KwaZulu-Natal, South Africa) for their help in interpreting the results and double-checking morphological identifications. We also thank the contributions of Didier Van den Spiegel (RMCA) for the elaboration of the original project, Brigitte Segers (RBINS) for the preparation of the tissue samples and Dr. Massimiliano Virgilio (RMCA) and Zoë Decorte (RMCA) for the DNA analysis.

## Author Contributions

**Conceptualization:** Gontran Sonet, Nathalie Smitz, Yves Samyn.

**Data curation:** Gontran Sonet, Yves Samyn.

**Formal analysis:** Gontran Sonet, Nathalie Smitz.

**Investigation:** Gontran Sonet, Nathalie Smitz, Carl Vangestel, Yves Samyn.

**Methodology:** Gontran Sonet, Nathalie Smitz, Carl Vangestel, Yves Samyn.

**Project administration:** Yves Samyn.

**Resources:** Gontran Sonet, Yves Samyn.

**Software:** Gontran Sonet.

**Supervision:** Gontran Sonet, Carl Vangestel, Yves Samyn.

**Validation:** Gontran Sonet, Nathalie Smitz, Yves Samyn.

**Visualization:** Gontran Sonet.

**Writing – original draft:** Gontran Sonet.

**Writing – review & editing:** Gontran Sonet, Nathalie Smitz, Carl Vangestel, Yves Samyn.

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
