## [Decision Letter · Decision Letter 0]

5 Jul 2022

PONE-D-22-16400DNA barcoding echinoderms from the East Coast of South Africa. The challenge to maintain DNA data connected with taxonomyPLOS ONE

Dear Dr. Sonet,

Thank you for submitting your manuscript to PLOS ONE. After careful consideration, we feel that it has merit but does not fully meet PLOS ONE’s publication criteria as it currently stands. Therefore, we invite you to submit a revised version of the manuscript that addresses the points raised during the review process.

We look forward to receiving your revised manuscript.

Kind regards,

Tzen-Yuh Chiang

Academic Editor

PLOS ONE

Journal Requirements:

6. We note that Figure 3 in your submission contain copyrighted images. All PLOS content is published under the Creative Commons Attribution License (CC BY 4.0), which means that the manuscript, images, and Supporting Information files will be freely available online, and any third party is permitted to access, download, copy, distribute, and use these materials in any way, even commercially, with proper attribution. For more information, see our copyright guidelines: http://journals.plos.org/plosone/s/licenses-and-copyright.

a. You may seek permission from the original copyright holder of Figure(s) [#] to publish the content specifically under the CC BY 4.0 license. 

Reviewers' comments:

Reviewer's Responses to Questions

**Comments to the Author**

1. Is the manuscript technically sound, and do the data support the conclusions?

Reviewer #1: Yes

2. Has the statistical analysis been performed appropriately and rigorously? 

Reviewer #1: I Don't Know

3. Have the authors made all data underlying the findings in their manuscript fully available?

Reviewer #1: No

4. Is the manuscript presented in an intelligible fashion and written in standard English?

Reviewer #1: Yes

5. Review Comments to the Author

Reviewer #1: Authors present quite standard procedures broadly utilized for barcoding diverse groups of animals. They also used four standard methods to delimit species based on acquired sequences, and compared them with classic morphological identifications. In Results and Discussion sections they discussed in details differences between those methods that predicted slightly different number of taxa. ?However, in my opinion the greatest weakened of the presented paper are illustrations. Authors discuss differences between algorithms, however, they didn't present results from each calculation, so it is impossible to compare them by a reader. These results should be at least attached as a supplementary materials. Moreover, on the Fig 3 authors wrote that trees were calculated using Neighbour Joining method. There is nothing about this method in M&M section. On the other hand there are no trees calculated using Bayes as mentioned i M&M. That should be definitely corrected or clarified. In general manuscript is well written, however, level of novelty of this research is limited.

6. PLOS authors have the option to publish the peer review history of their article (what does this mean?). If published, this will include your full peer review and any attached files.

Reviewer #1: No

---

## [Author Response · Author response to Decision Letter 0]

18 Aug 2022

Rebuttal letter (also attached in this submission as a doc file)

PONE-D-22-16400

DNA barcoding echinoderms from the East Coast of South Africa. The challenge to maintain DNA data connected with taxonomy

PLOS ONE

N.B. All replies to the comments of the reviewers start with ***

Review from the editor

***We read again the PLOS ONE's style requirements and adapted our manuscript accordingly.

***We adapted the ‘Funding Information’ section in order to match the Financial disclosure.

***ORCID ID was linked

4. Please include your full ethics statement in the ‘Methods’ section of your manuscript file. In your statement, please include the full name of the IRB or ethics committee who approved or waived your study, as well as whether or not you obtained informed written or verbal consent. If consent was waived for your study 

***We added the ethic statement in the material and methods: “All specimens studied here are invertebrates and were collected legally with written permissions from the regional and national South African authorities. The permit number of the expedition of 2016 is RES 2016-02 and was granted by the Department of Agriculture, Forestry and Fisheries (DAFF) and the Department of Environmental Affairs (DEA) of South Africa to Yves Samyn., please include this information in your statement as well.”

5. We note that Figure 1 in your submission contain [map/satellite] images which may be copyrighted. 

***We confirm that we have the permission to publish Figure 1 specifically under the CC BY 4.0 license. Indeed, the map was created using the software QGIS v. 3.22.6, the GEBCO Grid and Natural Earth. We added this information and the citation of all three sources in the caption of the figure and uploaded the terms of use of these tools as "Other"" file in the submission.

6. We note that Figure 3 in your submission contain copyrighted images. All PLOS content is published under the Creative Commons Attribution License (CC BY 4.0).

***We present a written permission from the copyright holder (a coauthor of this article) to publish these figures specifically under the CC BY 4.0 license (form uploaded as "Other"" file in the submission).

Reviewer's Comments to the Author

1. Is the manuscript technically sound, and do the data support the conclusions?

Reviewer #1: Yes

2. Has the statistical analysis been performed appropriately and rigorously? 

Reviewer #1: I Don't Know

3. Have the authors made all data underlying the findings in their manuscript fully available?

Reviewer #1: No

***All data supporting the findings related to this manuscript were uploaded a public repository (BOLD). As stated in the original submission, data will be released directly upon acceptance to publish this article in PONE. Based on the additional comments of the reviewer, we added all phylogenetic trees as supporting information (see below).

4. Is the manuscript presented in an intelligible fashion and written in standard English?

Reviewer #1: Yes

5. Review Comments to the Author

Reviewer #1: Authors present quite standard procedures broadly utilized for barcoding diverse groups of animals. They also used four standard methods to delimit species based on acquired sequences, and compared them with classic morphological identifications. In Results and Discussion sections they discussed in details differences between those methods that predicted slightly different number of taxa. However, in my opinion the greatest weakened of the presented paper are illustrations. Authors discuss differences between algorithms, however, they didn't present results from each calculation, so it is impossible to compare them by a reader. These results should be at least attached as a supplementary materials. 

***We totally agree with this comment. Illustrations are difficult to make for large data sets, and other large scale DNA barcoding studies already published also lack illustrative figures. Our Fig. 2 (general appreciation of the distances between the new sequences and those preexisting) and Fig. 3 (illustrated cases) are there to compensate. The different partitions obtained using the different algorithms were indeed not presented as raw data (they were included in Table 2 but we agree that this is not easy to consult. Hence, we added the putative species ID in the Table S1, which provides all info about all specimens and which can be easily reused by other researchers. The legend of the table was adapted: “S1 Table. Specimen information. Columns ABGD, GMYC, bPTP and RESL provide the putative species identifications for each method”.

Moreover, on the Fig 3 authors wrote that trees were calculated using Neighbour Joining method. There is nothing about this method in M&M section. 

***Indeed, we forgot to mention that “Neighbour joining trees based on p-distances were constructed using MEGA v7.0.26 (41) in order to illustrate distance among putative species.” This sentence was added in the Materials and Methods.

On the other hand there are no trees calculated using Bayes as mentioned i M&M. That should be definitely corrected or clarified. 

***We thank the reviewer for this observation. The original Materials and Methods section was indeed misleading in this regard. Bayesian inferences were generated in order to use them as an input for 2 of the 4 species delimitation methods (GMYC and bPTP), whereas Neighbour joining trees were generated to visualise the distances among the DNA sequences. In addition to the correction described above (concerning the neighbour joining trees), we also modified one sentence in the M&M as follow: “Phylogenetic trees were reconstructed through Bayesian inference (BI) and were used as an input for both GMYC and bPTP” to avoid any confusion. We also provide all 5 phylogenetic trees as Files S1-5 (one tree for each Class).

In general manuscript is well written, however, level of novelty of this research is limited.

***Thank you. “Level of novelty” is of course a relative appreciation. We agree that we do not provide a lot of firm conclusions, but we present data and analyses that will undoubtedly be useful for the taxonomy and systematics of echinoderms, and for future biodiversity inventories. 

6. PLOS authors have the option to publish the peer review history of their article (what does this mean?). If published, this will include your full peer review and any attached files.

Do you want your identity to be public for this peer review? For information about this choice, including consent withdrawal, please see our Privacy Policy.

Reviewer #1: No

***All three figures passed the PACE quality control.

---

## [Decision Letter · Decision Letter 1]

14 Sep 2022

PONE-D-22-16400R1DNA barcoding echinoderms from the East Coast of South Africa. The challenge to maintain DNA data connected with taxonomyPLOS ONE

Dear Dr. Sonet,

Thank you for submitting your manuscript to PLOS ONE. After careful consideration, we feel that it has merit but does not fully meet PLOS ONE’s publication criteria as it currently stands. Therefore, we invite you to submit a revised version of the manuscript that addresses the points raised during the review process.

We look forward to receiving your revised manuscript.

Kind regards,

Tzen-Yuh Chiang

Academic Editor

PLOS ONE

Journal Requirements:

Reviewers' comments:

Reviewer's Responses to Questions

**Comments to the Author**

1. If the authors have adequately addressed your comments raised in a previous round of review and you feel that this manuscript is now acceptable for publication, you may indicate that here to bypass the “Comments to the Author” section, enter your conflict of interest statement in the “Confidential to Editor” section, and submit your "Accept" recommendation.

Reviewer #2: All comments have been addressed

2. Is the manuscript technically sound, and do the data support the conclusions?

Reviewer #2: Yes

3. Has the statistical analysis been performed appropriately and rigorously? 

Reviewer #2: Yes

4. Have the authors made all data underlying the findings in their manuscript fully available?

Reviewer #2: Yes

5. Is the manuscript presented in an intelligible fashion and written in standard English?

Reviewer #2: Yes

6. Review Comments to the Author

Reviewer #2: The authors appear to have addressed all of the matters raised by previous reviews. They have done a good job of analysing and presenting a complex mix of existing and new DNA barcode data in a poorly studied group. It is often very difficult to combine new barcode data with existing data which not infrequently contains sequences attributed to incorrect species.

The authors point out the limitations in their dataset and highlight areas that need further clarification in future.

I make only minor comments on the methods which the authors should address in the final version of the manuscript, nevertheless I reccomend the manuscript be accepted for publication:

Lines 161-163: NJ trees were based on p-distances. Building trees based on uncorrected distances has not been acceptable practice since model-based distance corrections became commonplace in the 1980s. While Bayesian or ML trees would be vastly preferable, NJ trees are acceptable for barcode analyses if based on corrected distances. MEGA has many options for distance correction. BOLD uses K2P-distances by default so that would be a good choice. Regardless, if the NJ trees are never shown then what is their purpose? I suggest deleting all reference to NJ trees and using the Bayesian trees.

Lines 163-164: On what date was the search function used on BOLD to retrieve sequences? This information is vital for reproducibility of methods.

Lines 181-183: Although BOLD produces BINs based on the RESL algorithm, it is not correct to equate BINs to the results of RESL analysis because BOLD uses additional criteria to assign sequences to BINs, including data quality standards (sequence length, etc.). It is common to have only a subset of sequences in a dataset assigned to BINs. However, it is possible to perform RESL analysis on BOLD for all sequences in your dataset, regardless of whether BOLD assigns them to a BIN. Besides which, BINs are not assigned immediately but only periodically as BOLD analyses new sequences for BIN assignment. So, did the authors perform RESL analyses or not? Were all their sequences assigned to BINs? The authors need to clarify exactly what they did.

7. PLOS authors have the option to publish the peer review history of their article (what does this mean?). If published, this will include your full peer review and any attached files.

Reviewer #2: **Yes: **Andrew Mitchell

---

## [Author Response · Author response to Decision Letter 1]

19 Sep 2022

Rebuttal letter

 Nota bene: 

Responses to the reviewer are in bold and follow the signs “>>>”. 

Actions taken in the manuscript follow the signs “->”. 

Reviewers' comments:

Reviewer's Responses to Questions

Comments to the Author

Reviewer #2: All comments have been addressed

2. Is the manuscript technically sound, and do the data support the conclusions?

Reviewer #2: Yes

3. Has the statistical analysis been performed appropriately and rigorously? 

Reviewer #2: Yes

4. Have the authors made all data underlying the findings in their manuscript fully available?

Reviewer #2: Yes

5. Is the manuscript presented in an intelligible fashion and written in standard English?

Reviewer #2: Yes

6. Review Comments to the Author

Reviewer #2: The authors appear to have addressed all of the matters raised by previous reviews. They have done a good job of analysing and presenting a complex mix of existing and new DNA barcode data in a poorly studied group. It is often very difficult to combine new barcode data with existing data which not infrequently contains sequences attributed to incorrect species.

>>> Thank you. Yes, we agree that it represented a major part of the effort involved in this manuscript

The authors point out the limitations in their dataset and highlight areas that need further clarification in future.

>>> Yes, this is important of course

I make only minor comments on the methods which the authors should address in the final version of the manuscript, nevertheless I reccomend the manuscript be accepted for publication:

Lines 161-163: NJ trees were based on p-distances. Building trees based on uncorrected distances has not been acceptable practice since model-based distance corrections became commonplace in the 1980s. While Bayesian or ML trees would be vastly preferable, NJ trees are acceptable for barcode analyses if based on corrected distances. MEGA has many options for distance correction. BOLD uses K2P-distances by default so that would be a good choice. Regardless, if the NJ trees are never shown then what is their purpose? I suggest deleting all reference to NJ trees and using the Bayesian trees.

>>> We totally agree that uncorrected distances are not recommended for phylogenetic tree reconstructions. That is why we used Bayesian inference to reconstruct the ultrametric phylogenetic trees that were used for tree-based species delimitations. Here we simply used uncorrected distances to illustrate the proportion of substitutions observed among some putative species in Fig. 3. In our opinion, these trees are good for illustration (because they are without any a priori model-based distances) if they are introduced correctly to avoid risks of misinterpretation. We understand that this is debatable and apply the changes suggested by the reviewer.

  We remove the reference to the NJ tree and Fig. 3. which are only there to illustrate a content, which is already fully described in the main text and in the tables.

Lines 163-164: On what date was the search function used on BOLD to retrieve sequences? This information is vital for reproducibility of methods.

>>> We agree that it is of primary importance and the information was already in reference ‘25’ cited on line 163. 

  To make it more visible, we repeat it in the main text by adding “(…) on the 24 May 2022”.

Lines 181-183: Although BOLD produces BINs based on the RESL algorithm, it is not correct to equate BINs to the results of RESL analysis because BOLD uses additional criteria to assign sequences to BINs, including data quality standards (sequence length, etc.). It is common to have only a subset of sequences in a dataset assigned to BINs. However, it is possible to perform RESL analysis on BOLD for all sequences in your dataset, regardless of whether BOLD assigns them to a BIN. Besides which, BINs are not assigned immediately but only periodically as BOLD analyses new sequences for BIN assignment. So, did the authors perform RESL analyses or not? Were all their sequences assigned to BINs? The authors need to clarify exactly what they did.

>>> We thank the reviewer for these precisions. We did not perform RESL analyses ourselves. We indeed used the BIN IDs when they became available on BOLD. For a few sequences without a BIN, we followed the advice of the support team of BOLD (email of 12 June 2020) who advised to use the “Cluster Sequences tool” in the workbench to group COI sequences into OTUs. If the other sequences in the OTU have a BIN, it is likely that those without a BIN belong to that same one, but may not be included in it for some quality check reasons, otherwise they would have been assigned to that BIN.

  In the text, we replaced “RESL” by “BIN”. The way we obtained the BINs was already clarified in lines 182-187 (but the “RESL” abbreviation was inded wrongly used). It reads now: “The BINs were obtained on BOLD. (…). When the sequence overlap with the BOLD sequences was lower than 300 bp and a BIN was not attributed directly by BOLD, the BIN of the best match above 99% similarity was assigned to the record.”

---

## [Editor Report · Decision Letter 2]

23 Sep 2022

DNA barcoding echinoderms from the East Coast of South Africa. The challenge to maintain DNA data connected with taxonomy

PONE-D-22-16400R2

Dear Dr. Sonet,

We’re pleased to inform you that your manuscript has been judged scientifically suitable for publication and will be formally accepted for publication once it meets all outstanding technical requirements.

Kind regards,

Tzen-Yuh Chiang

Academic Editor

PLOS ONE
---

## [Editor Report · Acceptance letter]

30 Sep 2022

PONE-D-22-16400R2 

DNA barcoding echinoderms from the East Coast of South Africa. The challenge to maintain DNA data connected with taxonomy 

Dear Dr. Sonet:

I'm pleased to inform you that your manuscript has been deemed suitable for publication in PLOS ONE. Congratulations! Your manuscript is now with our production department. 

Kind regards, 

on behalf of

Dr. Tzen-Yuh Chiang 

Academic Editor

PLOS ONE